# Personalization-based deep hybrid E-learning model for online course recommendation system

Subha S[1], Baghavathi Priya Sankaralingam[2], Anitha Gurusamy[2], Sountharrajan Sehar[2] and Durga Prasad Bavirisetti[3]

[1] Department of Computer Science and Engineering, Rajalakshmi Institute of Technology, Chennai, Tamil Nadu, India

[2] Department of Computer Science and Engineering, Amrita School of Computing, Amrita Vishwa Vidyapeetham, Chennai, Tamil Nadu, India

[3] Department of Computer Science, Norwegian University of Science and Technology (NTNU), Trondheim, Sor Trondlog, Norway



## ABSTRACT

Deep learning, a subset of artificial intelligence, gives easy way for the analytical and physical tasks to be done automatically. There is a less necessity for human intervention while performing these tasks. Deep hybrid learning is a blended approach to combine machine learning with deep learning. A hybrid deep learning (HDL) model using convolutional neural network (CNN), residual network (ResNet) and long short term memory (LSTM) is proposed for better course selection of the enrolled candidates in an online learning platform. In this work, a hybrid framework that facilitates the analysis and design of a recommendation system for course selection is developed. A student's schedule for the next course should consist of classes in which the student has shown interest. For universities to schedule classes optimally, they need to know what courses each student wants to take before each course begins. The proposed recommendation system selects the most appropriate course that can encourage students to base their selection on informed decision making. This system will enable learners to obtain the correct choices of courses to be studied.

## INTRODUCTION

The recommendation system is an emerging science brought into existence by the conjunction of several established paradigms, education, medicine, artificial intelligence, computer science, and the many derivatives of these fields. By utilizing machine learning algorithms, numerous investigations and research projects for course selection have been conducted. Deep learning algorithms are now being employed to help users select a reliable course for their skill improvement. E-learning is a method of training and education that uses digital resources. Through any of the well-known learning management systems (LMS), such as Moodle, Coursera, NPTEL, *etc.*, users can learn at any time, anywhere. Selecting courses from an LMS is currently a difficult decision for users. As there is an

Corresponding author
Durga Prasad Bavirisetti,
durga.bavirisetti@ntnu.no

enormous amount of multi-disciplinary courses, sometimes, users are unable to choose the right choices of courses based on their interests and specialization. Initially, a person can start with a low-complexity course so that he or she can get deep knowledge in the fundamental concepts of a technology/domain. If he completes the course, he is further recommended to choose the medium-level complexity course in the same discipline. In continuation to, the final stage is choosing the high-level complexity course. This will enable the learner to get very strong knowledge in the particular field as he has completed all three levels of courses. A hybrid deep learning model is developed by utilizing the architectures of convolutional neural network (CNN), residual network (ResNet) and long short term memory (LSTM) for efficient course selection. Since each problem may be solved using various methods, there must also be an infinite number of available methods. The previous studies most relevant to the selection of courses for learners are highlighted in this section.

There are different resources, such as e-learning log files, academic data from students, and virtual courses, from which educational data can be obtained. It is still a challenging factor in the educational field to predict student marks (*Turabieh, 2019*). During the COVID period, video conferencing platforms and learning management systems are being adopted and they are used as online learning environments (OLEs). It was suggested that learners' behavior in OLEs can be predicted using effective methods and they can be made available as supportive tools to educators (*Dias et al., 2020*).

Since the previous decade, the field of recommendation systems has expanded at a faster pace. Due to its enormous importance in this era, much work is being done in this area. Similar suggestions cannot be made to users in e-learning, despite their shared interests. The suggestion heavily depends on the different characteristics of the learners. It has been determined that because each learner differs in terms of prior information, learner history, learning style, and learning objectives, all learners cannot be held to the same standards for recommendations (*Chaudhary & Gupta, 2017*).

A collaborative recommender system is described that considers both the similarities and the differences in the interests of individual users. An association rule mining method is used as the fundamental approach so that the patterns between the different classes can be found (*Al-Badarenah & Alsakran, 2016*).

A collaborative recommendation system is designed by utilizing natural language processing and data mining techniques. These techniques help in the conversion of information from a format that is readable by humans to one that is readable by machines. The system allows graduate students to select subjects that are appropriate for their level of expertise (*Naren, Banu & Lohavani, 2020*). A recommendation system is described for selecting university elective courses. It is based on the degree to which the student's individual course templates are similar to one another.

This study makes use of two well-known algorithms, the alternating least squares and Pearson correlation coefficient, on a dataset consisting of academic records from students attending a university (*Bhumichitr et al., 2017*).

An interactive course recommendation system, CourseQ enables students to discover courses *via* the use of an innovative visual interface. It increases transparency of course

suggestions as well as the level of user pleasure they provide (*Ma et al., 2021*). A hybrid recommender system with collaborative filtering and content-based filtering models using information on students and courses is proposed that generates consistent suggestions by making use of explicit and implicit data rather than predetermined association criteria (*Alper, Okyay & Nihat, 2021*). The content-based filtering algorithms make use of textual data, which are then transformed into feature vectors *via* the use of natural language processing techniques. Different ensembling strategies are used for prediction.

Based on score prediction, a course recommendation system is developed where the score that each student will obtain on the optional course is reliably predicted by a cross-user-domain collaborative filtering algorithm (*Huang et al., 2019*). The score distribution of the senior students who are most comparable to one another are used to achieve this. A recommendation system by makes use of a deep neural network is described in such a way that the suggestions are produced by gathering pertinent details as characteristics and assigning weights to them. The frequency of nodes and hidden layers has been updated by the use of feed forwarding and backpropagation information, and the neural network is formed automatically through the use of a great number of modified hidden layers (*Anupama & Elayidom, 2022*). The automatic construction of convolutional neural network topologies using neuro-evolution has been examined, and a unique method based on the Artificial Bee Colony and the Grey Wolf Optimizer has also been developed (*Karthiga, Shanthi & Sountharrajan, 2022*).

To maximize the degree of curricular system support, a mathematical model is developed that uses a two-stage improved genetic firefly algorithm. It is based on the distribution of academic work across the semesters and the Washington Accord graduate attribute (*Jiang & Xiao, 2020*).

Another recommendation algorithm takes into account a student's profile as well as the curricular requirements of the course they are enrolled in. Integer linear programming and graph-based heuristics are used to find a solution for course selection (*Morrow, Hurson & Sarvestani, 2020*).

A recommender system is intended to help the investors of the stock market find potential possibilities for-profit and to aid in improving their grasp of finding pertinent details from stock price data (*Nair et al., 2017*).

In an improved collaborative filtering for course selection, the user's implicit behavior is the foundation of this collaborative filtering system. The behavioral data are mined, and then using the collaborative filtering algorithm based on items, an intelligent suggestion is carried out (*Zhao & Pan, 2021*).

Using learning analytics, data from students' learning activities can be collected and analyzed. It is proposed that a particular computational model can be utilized to identify and give priority to students who were at risk of failing classes or dropping out altogether. The results of predictive modeling can be used to inform later interventional measures (*Wong, 2017*). Using LSTM network architecture, efficient online learning algorithms were introduced and they make use of the input and output covariance information. Weight matrices were assigned to the input and output covariance matrices and were learned sequentially (*Mirza, Kerpicci & Kozat, 2020*). A hybrid teaching mode is proposed that

introduces a support vector machine to predict future learning performance. Cluster analysis is used to analyze the learner's characteristics, and the predicted results are matched with the hybrid mode for continuing the offline teaching process (*Liang & Nie, 2020*).

In another study, a deep neural network model, namely, the attention-based bidirectional long short-term memory (BiLSTM) network was examined to predict student performance (grades) from historical data using advanced feature classification and prediction (*Yousafzai et al., 2021*). A recommender system that helps the appropriate users see the correct ads. It puts this concept into practice using two cooperative algorithms CNN and LSTM (*Soundappan et al., 2015*).

A hybrid e-learning model was suggested as a way to apply distance learning in Iraqi universities. Seventy-five individuals from the University of Technology in Iraq took part in this study (*Alani & Othman, 2014*). The CNN and LSTM algorithms are outperformed in large volume of data with less computational cost (*Anitha & Priya, 2022*). Distinct trade-offs between large-scale and small-scale learning systems using various gradient optimization algorithms were discussed in a article. Gradient descent, stochastic gradient descent, and second order stochastic gradient descent were analyzed and it was concluded that the stochastic algorithms yield the best generalization performance (*Bottou & Bousquet, 2007*). With logistic outputs and MSE training for the OCR challenge, LSTM networks were unable to converge to low error rate solutions; softmax training produced the lowest error rates overall. In every experiment, standard LSTM networks without peephole connections had the best performance. The performance of the LSTM is gradually dependent on learning rates. Least square training falls short of softmax training (*Breuel, 2015*).

A Prediction Model for Course Selection (PMCS) system was developed using a Random Forest algorithm (*Subha & Priya, 2023*). As the number of decision trees in the Random Forest classifier is increased, the performance of the PMCS system improves. It is observed that the Random Forest classifier with 75 trees performs well when compared to other decision trees. Although RMSProp and ADAM are still very well-liked neural net training techniques, it is still unknown how well they converge theoretically. We demonstrate the criticality of these adaptive gradient algorithms for smooth non-convex objectives and offer limitations on the running time (*De, Mukherjee & Ullah, 2018*). The restricted fundamental enrolling features of Taiwan's current course selection systems prevent them from predicting performance or advising students on how to arrange their courses depending on their learning circumstances. Therefore, students select courses with a lack of awareness about it and they could not prepare for a study schedule. The learning curve is defined by analyzing the required factors and students with different background was found out initially. Then, a recommendation system is developed based on the students' grade and their learning curve (*Wu & Wu, 2020*). A virtual and intelligent agent-based recommendation is proposed which needs user profiles and preferences. User rating results are improved by applying machine learning techniques (*Shahbazi & Byun, 2022*). Existing research works did not concentrate on the hybrid model which combines the features of CNN, ResNet, and LSTM. They have focused on individual models only.

Though there are some considerable merits in a few individual models, the hybrid model efficiently outperforms those models.

In this work, the proposed recommendation system develops a hybrid approach utilizing three deep learning architectures: CNN, ResNet, and LSTM. The proposed HDL approach selects the best optimal course for the learners. The remaining sections are summarized as follows: "Materials and Methods" gives an overview of the proposed recommendation system with a brief discussion of each module. The next section goes deeper into the recommendation system's metrics and evaluates its effectiveness by using a student dataset having 750 records. "Conclusion" summarizes the findings and ideas for future work.

## MATERIALS AND METHODS

The framework used for recommending courses *via* the utilization of the hybrid technique is shown in Fig. 1. Considered to be a pattern recognition system, the recommendation system comprised three different deep learning architectures, CNN, ResNet, and LSTM, as base models. The designs of individual architecture for the proposed HDL system are discussed in the following subsections.

### Preprocessing

When training a neural network, such as a CNN, ResNet, and LSTM, the input data for the classification system must be scaled. The unscaled data slows down the learning and convergence of the network as the range of values is high. They may even prevent the network from effectively learning the problem when it is fitted. Normalization and standardization are two approaches to scaling the input data.

When applying standardization, it assumes that the input data follow a Gaussian distribution, often known as a bell curve, with a mean and standard deviation that behave predictably. It is still possible to standardize data even if this assumption is not realized; however, the results you get may not be accurate. Sometimes it will be difficult to interpret the data due to loss of information. Student datasets used in the study contain different grade levels of the students. Also, the grades may be skewed towards the left, which means that there are more students with higher grades. Here the data is not evenly distributed around the mean and this is one of the major reasons for not following Gaussian distribution. Also, the data must be independent for Gaussian distribution. However the student dataset cannot be independent as there is a way to correlation of grades among the students. Therefore, normalization is employed in this study.

The process of normalization involves rescaling the data from their original range to a new range in which all of the values are contained between 0 and 1. The ability to know the lowest and maximum observable values, or to have an accurate ability to estimate them, is necessary for normalization. Deep neural networks are enhanced in accuracy and performance by stacking additional layers to solve complicated problems. The idea behind adding additional layers is that they will eventually acquire more complex features. The following deep learning models are analyzed and compared to build an efficient recommendation system.

**Peer**J Computer Science

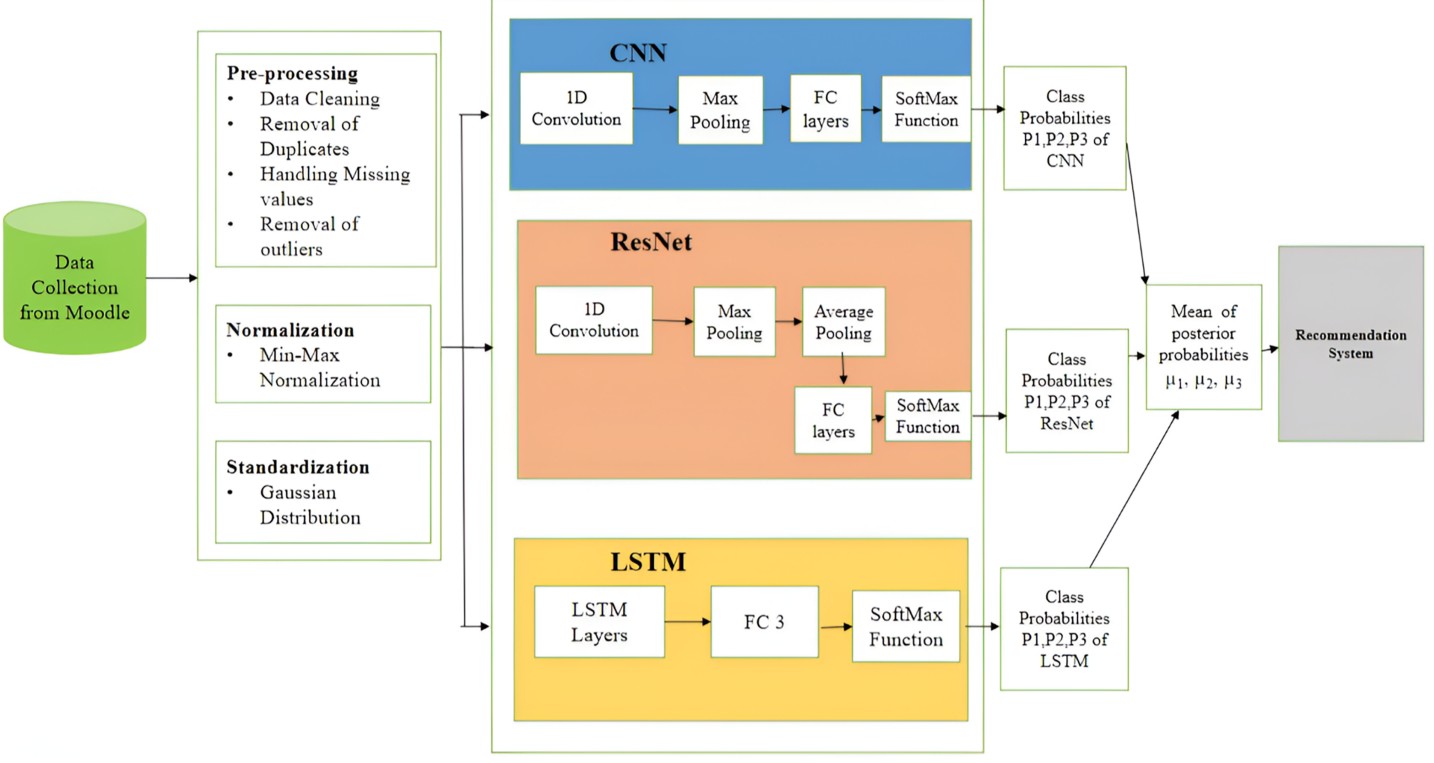

**Figure 1 Architecture diagram of the hybrid recommendation system.**     

## Recommendation system with CNN architecture

CNNs are built from two basic components, which are referred to as the convolutional layers and the pooling layers. Even though they are straightforward, there are many ways to organize these layers in any computer vision system. It is possible to construct incredibly deep convolutional neural networks by making use of common patterns for configuring these layers as well as architectural improvements. Figure 2 shows the proposed system with CNN architecture for the recommendation system.

In Fig. 2, ODC represents one-dimensional convolution, MP represents max pooling with stride (S) of 2, and FC represents fully connected layers. The ODC by a kernel of size *px* is defined in Eq. (1).

$$y_i = \sum_{i=-p}^{p} X_{j-i} w_i \tag{1}$$

where *x* is the input data and *w* is the weight of the kernel. In MP, the word "stride" refers to the total number of shifts that occur throughout the vectors. The convolution filter is relocated to two values in the input vector when using a stride value of 2.

The convolution and MP are applied repeatedly as per the design of the recommendation system with CNN architecture in Fig. 2, and the extracted deep features reach the FC layer at last. The proposed design has three FC layers, and the classification

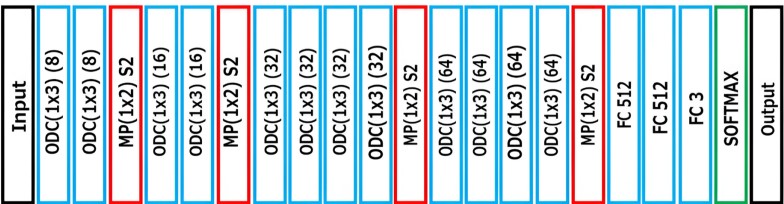

**Figure 2 Recommendation system with CNN architecture.**

takes place at the final FC layer. The FC layer consists of a neural network architecture for prediction. It is motivated by the biological brain system's process of learning, which involves the memorization and recognition of patterns and correlations of data gleaned from prior knowledge and the compilation of information gleaned from past experiences to forecast a certain event.

The neural network architecture has no less than three layers, namely, an input layer, a hidden layer, and an output layer. Layers are built of neurons (nodes) that are coupled to one another. One of the most effective algorithms for making predictions or performing classifications is known as the feed-forward neural network. The difference that exists between the input and the weights is referred to as the error, and it is used to modify the weights to decrease the error that occurs during the prediction process. This helps to establish the most accurate output that can be accurately predicted. The proposed system with CNN architecture uses cross-entropy loss, which is defined in Eq. (2).

$$Error\ or\ loss = -\sum_{i=2}^{n} GT_i\ log(SMP_i) \tag{2}$$

where $SMP_i$ represents the softmax probability and $GT_i$ is the ground truth label. The other parameter settings are presented in Table 1.

## The algorithm for the recommendation system based on the CNN model

1. The results after pre-processing are considered as the input data.

2. Perform one-dimensional convolution with the layer values of $1 \times 3$ (8) twice.

3. Perform max pooling with stride $(1 \times 2)$ S2.

4. Repeat steps 2 and 3 for the layer values of $1 \times 3$ (16) twice, $1 \times 3$ (32) twice and $1 \times 3$ (64) twice.

5. Send the final values to the fully connected layer with sizes 512 and 3.

6. Apply the softmax function to get the corrected output.

The major advantages of using CNN are reduced overfitting, improved accuracy, and computational efficiency. Additionally, CNNs can learn useful features directly from the input data. However, the drawback of the CNN model is that CNNs require a large amount of labeled data to train effectively, which can be a challenge for tasks where labeled data are scarce or expensive to obtain. To overcome this challenge, the ResNet model has been applied to recommendation systems.

**Table 1 Training parameters for the system with CNN architecture.**

| Parameters | Value/type | Parameters | Value/type |
|---|---|---|---|
| Epochs | 50 | Loss Function | Cross entropy |
| Momentum | 0.9 | Optimizer algorithms | Stochastic descent gradient, Adam optimization and root mean square propagation. |
| Activation (input layers) | Rectified linear | Output layer | Softmax |

## Recommendation system with residual network

Following the development of CNN, ResNet was established. In a deep neural network, more layers may be added to increase its accuracy and performance, which helps to tackle difficult tasks. The working hypothesis is that each successive layer would gradually learn more features for better performance. Figure 3 shows the proposed system with ResNet architecture for the recommendation system.

A specific type of neural network called a residual network (ResNet) skips some layers in between and has a direct link. The core of residual blocks is known as the "skip connection." Next, this term passes through the activation function, f(), and the output H (x) is defined in Eq. (3).

$$H(x) = f(wx + b) \text{ or } H(x) = f(x) \tag{3}$$

Equation (4) represents the output with a skip connection.

$$H(x) = f(x) + x \tag{4}$$

Using this additional short-cut path for the gradient to flow through, ResNet's skip connections address the issue of vanishing gradients in deep neural networks. In this case, the higher layer will function at least as well as the lower layer, if not better.

## Algorithm for the recommendation system based on the ResNet model

1. Get the input values from the training dataset.
    2. Perform one-dimensional convolution with the layer values of $1 \times 5$ (64) one time.
    3. Perform max pooling with stride $(1 \times 2)$ S2.
    4. Perform one-dimensional convolution with the layer values of $1 \times 3$ (8) twice.
    5. Repeat step 4 for the layer values of $1 \times 3$ (16) twice, $1 \times 3$ (32) four times and $1 \times 3$ (64) four times.
    6. Apply average pooling and send the final values to the fully connected layer FC-3.
    7. Apply the softmax activation function to obtain the desired output.

ResNet does not have more training errors. Additionally, there is an improved accuracy and an increase in efficiency. Much deeper networks can be trained using ResNet. However, some of the pitfalls in using ResNet are the difficulty in initializing the

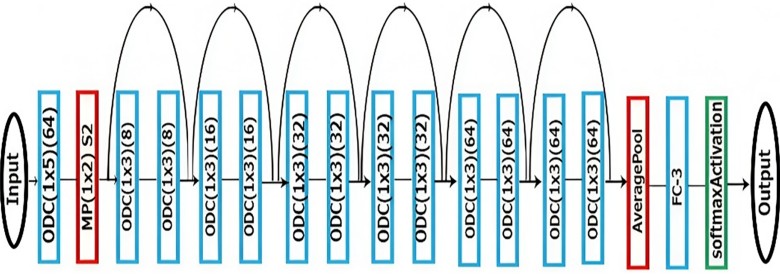

**Figure 3** **Recommendation system with ResNet architecture.**

parameters for ResNet and complexity, computational expense, and overfitting. To overcome the complexity, another deep learning model, namely, long short-term memory (LSTM), has been introduced and can be used in recommendation systems. LSTMs can model complex relationships between input and output sequences and are suitable for tasks where the output depends on long-term dependencies in the input data.

## Recommendation system with LSTM architecture

LSTM is a recurrent neural network (RNN) built for solving complex problems such as in predicting sequences. An RNN is a looped feed-forward network. Figure 4 shows the proposed system with LSTM architecture.

The LSTM network's basic computational building element is referred to as the memory cell, memory block, or simply cell. The word "neuron" is so often used to refer to the computational unit of neural networks that it is also frequently used to refer to the LSTM memory cell. The weights and gates are the constituent parts of LSTM cells.

A memory cell includes weight parameters for the input, output, and an internal state. The internal state is built throughout the cell's lifetime. The gates are essential components of the memory cell. These functions, like the others, are weighted, and they further influence the flow of the cell's information. To update the state of the internal device, an input gate and a forget gate are used. The output gate is a final limiter on what the cell outputs. The recommendation system with LSTM architecture uses stacked LSTM that makes the model deeper to provide more accurate prediction results.

The representations of the input gate, false gate, and output gate of LSTM are given in Eqs. (5)–(7) respectively.

$$i_t = \widetilde{A}(w_i[h_{t-1}, x_t] + b_i) \tag{5}$$

$$f_t = \widetilde{A}(w_f[h_{t-1}, x_t] + b_f) \tag{6}$$

$$o_t = \widetilde{A}(w_o[h_{t-1}, x_t] + b_o) \tag{7}$$

where $i_t$ represents the input gate, $f_t$ represents the forget gate, it represents the output gate, $\widetilde{A}$ represents the sigmoid function, $w_x$ denotes the weight for the respective gate(x) neurons, $h_{t-1}$ denotes the output of the previous LSTM block (at timestamp t-1), tx denotes the input at the current timestamp and $b_x$ denote biases for the respective gates(x).

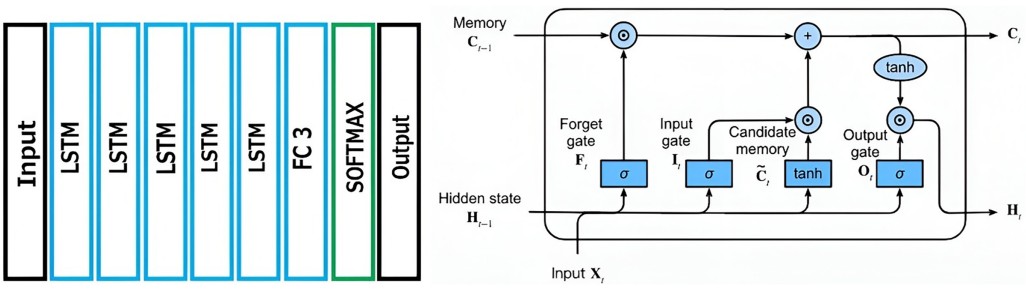

**Figure 4** Recommendation system with LSTM architecture.

## Algorithm for the recommendation system based on the LSTM model

1. Get the input values from the training dataset.
   2. Pass the values to the LSTM layers.
   3. Then, the values are fed to the fully connected layer FC 3.
   4. Apply the softmax function to reach the output.

LSTMs are designed to handle sequential data, which is well suited for time series prediction. They can handle the vanishing gradient problem that occurs in RNNs on long sequences. There is an improved performance when compared to RNNs and other neural networks on a variety of sequential data tasks. However, the challenge is still the difficulty in initialization and difficulty in interpreting results.

## Activation function in the deep learning models

In all the above models, the softmax activation function is used. The Softmax function can be applied to multiclass problems, and the output probability range will be from 0 to 1. The total of all probabilities will sum to 1. Therefore, this function is chosen to be used in all the architectures. The sigmoid activation function can be applied to binary classification methods only, and it is most prone to the vanishing gradient problem. Additionally, the output is not zero-centered For better computation performance and to avoid the vanishing gradient issue, a different activation function called the rectified linear unit (ReLU) can be used in the hidden layer. These are the main reasons for choosing only the softmax function in the individual models and the hybrid model.

If the above models function separately, they do not produce optimal results. Additionally, to overcome the computational complexity problem that occurred in the above three architectures, there is a need to develop a hybrid model that combines the features of CNN, ResNet and LSTM.

## Challenges in other deep learning models

There are few challenges in many other deep learning architecture, such as gated recurrent units (GRUs), generative adversarial networks (GANs), and autoencoders (AEs). GRUs are difficult to interpret, as the internal workings of the model are not as transparent as traditional machine learning models. GANs can be sensitive to hyperparameters such as the size of the hidden layers, the type of activation function used, and the optimization

algorithm used, which can affect the performance of the model. Autoencoders are designed to reconstruct the input data rather than learn more complex relationships between the inputs and outputs. Additionally, they are designed mainly for unsupervised learning tasks, and they cannot handle noisy data.

## Hybrid system

The proposed work is aimed at developing a hybrid model that combines the three deep learning models namely CNN, ResNet, and LSTM. A hybrid model not only provides better robustness and improved accuracy; it also reduces computational complexity. An efficient course selection can be done and a better course recommendation can be given to the enrolled users with the aid of the hybrid system.

The final layer of the proposed system uses architectures such as CNN, ResNet and LSTM is the softmax layer where the recommendation takes place. It transforms the scores to a normalized probability distribution, which may then be shown to a user or utilized as input by other systems. It is defined in Eq. (8) as

$$SoftMax(y_n) = \sum_m \frac{e^{y_n}}{e^{y_m}} \tag{8}$$

where $in$ and $m$ are the output of $the\ n^{th}$ layer and the number of classes, respectively. To hybridize the given architectures, a score (probability) fusion technique is employed. In the probability fusion module, probabilities from multiple sources namely CNN, ResNet and LSTM are aggregated and given to the hybrid model for recommendation, thereby improving the overall prediction. Figure 5 shows the fusion module for the proposed recommendation system.

In Fig. 5, P1, P2, and P3 represent the probability of class 1 to class 3, respectively. The obtained probabilities of each architecture are combined using several approaches to generate a new score, which is then used in the decision module to make the final decision. This method's effectiveness and reliability are directly correlated to the quantity and quality of the input data used in the training phase. In addition, the scores from each architecture do not have to be consistent with one another; hence, the normalization of scores is not needed. This proposed work uses the max rule for the final decision by estimating the mean of the posterior probabilities ($\mu1$, $\mu2$, and $\mu3$) by the maximum value. Commonly used probability fusion techniques are weighted averaging, max/min fusion, softmax fusion, logit fusion, and rank-based fusion. Among these techniques, multiple learning algorithms are used in voting-based ensemble approaches, which strengthen the classification model. Weighted voting-based ensemble methods offer a more flexible and fine-grained manner to anticipate exact output classes when compared to unweighted (majority) voting-based ensemble methods. In this work, the weighted voting method is applied in the probability fusion module to get the mean of the posterior probabilities thereby aiming to get the actual output class. Using the weighted averaging method, the probabilities from each separate model are combined. The average probability of each model is calculated and are taken together to be used in the probability fusion module. The

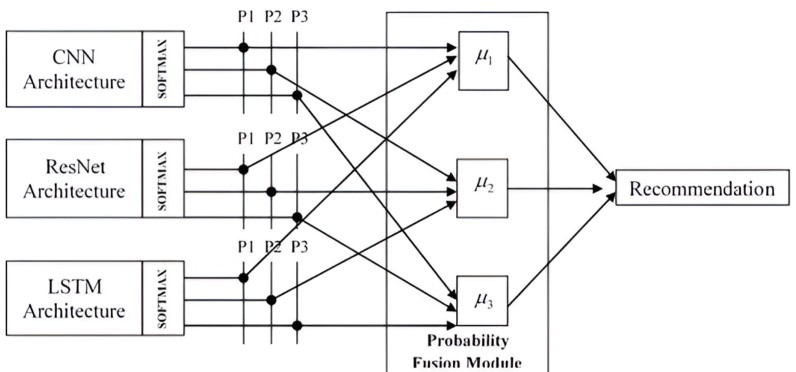

**Figure 5 Probability fusion module of the hybrid recommendation system.**

mean value of the posterior probabilities is calculated from the probability values of three individual models namely CNN, ResNet, and LSTM.

## Algorithm for the recommendation system using the hybrid model

1. Do follow the steps mentioned in the algorithm of the CNN model.
2. Do follow the steps mentioned in the algorithm of the ResNet model.
3. Do follow the steps mentioned in the algorithm of the LSTM model.
4. Obtain the final results of the softmax function of each model.
5. From the softmax function of CNN, ResNet, and LSTM, probability values P1, P2 and P3, which represents classes 1 to 3, respectively, is obtained for each architecture.
6. Apply the probability values to the probability fusion module.
7. Use the weighted voting method in the probability fusion module to obtain the mean of the posterior probabilities.
8. In the weighted average voting method, each probability estimate is multiplied by a wight that represents the reliability of that estimate.
9. The final fused probability is obtained by summing up all the weighted probabilities.
10. The mean value $\mu1$ is calculated from the p1 values of CNN, ResNet, and LSTM.
11. Similarly, the other two mean values are calculated from the p2 and p3 values of all three individual models.
12. The resultant posterior probability values are recommended for the hybrid model.

## Optimization algorithms used in the recommendation system

The performance of the HDL system and its networks are analyzed using three different optimization algorithms: stochastic gradient descent (SGD), Adam optimization (AO) and root mean square propagation (RMSProp). Gradient descent is formally classified as a first-orderoptimization algorithm because it makes explicit use of the $1^{st}$ order derivative of the target $f$ (). The first stage in making a downhill movement is to compute how far it should move in the input space. This is done by multiplying the gradient by the step size, which is referred to as $\alpha$ or the learning rate. After that, this is subtracted from the current

point, which ensures that we are moving in the opposite direction as the gradient. It is defined by

$$x(t) = x(t-1) - \alpha \times f'(x(t-1)) \tag{9}$$

If the step size is not large enough, there will be very little movement in the region being searched, which will cause the process to take a very lengthy period. If the step size is too large, the search may miss the optimal solution as it bounces about the search space RMSProp is an extension of SGD that improves the capability of SGD optimization. It is created to test the hypothesis that different parameters in the search space need different $\alpha$ values. The partial derivative is used to advance along a dimension once a calculated $\alpha$ has been determined for that dimension. For each additional dimension of the search space, this procedure is repeated. The custom step size is defined by

$$\text{cust\_}\alpha = \frac{\alpha}{(10^{-8} + \text{RMS(s)})} \tag{10}$$

where $s$ is the sum of the squared partial derivatives of the input variable that have been found throughout the search. Adam optimization is the combination of RMSProp and adaptive gradient. The former performs well for noisy data, whereas the latter performs well on computer vision and natural language problems. Adam utilizes the average of the second moments of the gradients in addition to the average of the first moment (the mean) when adjusting the learning rates of the parameters (the uncentered variance). It also computes an exponential moving average of the gradient and the squared gradient, with decay rates determined by the parameters $\beta_1$ and $\beta_2$

$$x(t) = x(t-1) \frac{a(t) \times m(t)}{\sqrt{v(t)} + 10^{-8}} \tag{11}$$

where

$$\alpha(t) = \alpha \times \frac{\sqrt{1 - \beta_2(t)}}{1 - \beta_1(t)} \tag{12}$$

$$m(t) = \beta_1 \times m(t-1) + (1 - \beta_1) \times f'(x(t-1)) \tag{13}$$

$$v(t) = \beta \times v(t-1) + (1 - \beta) \times f'(x(t-1)^2) \tag{14}$$

## RESULTS

The true effectiveness of the recommendation system is evaluated using real-time samples in this section. Samples are obtained from the learners of an engineering college in Tamil Nadu with the help of the Moodle learning management system. Student dataset consists of the following attributes: S.No., Student-Id, Gender, Region, Age, Course Code, Course Duration, Course Label, Course Presentation, Date registered, Course Start Date, Total credits, Average course feedback, No. of previous attempts, Assessment_id, Assessment type, Date of submission, No. of days viewed, No. of clicks per day, Total no. of clicks, and Total no. of exercises completed.

**Table 2 CNN based recommendation system performance.**

| Combination | Confusion matrix | | | | Parameters | | | | Performance indexes | | | |
|---|---|---|---|---|---|---|---|---|---|---|---|---|
| | Category | H | M | L | TP | FP | TN | FN | Acc | Re | Pr | F1 |
| CNN + SGD | High | 185 | 34 | 31 | 185 | 56 | 444 | 65 | 83.87 | 74.00 | 88.80 | 80.73 |
| | Medium | 29 | 192 | 29 | 192 | 62 | 438 | 58 | 84.00 | 76.80 | 87.60 | 81.85 |
| | Low | 27 | 28 | 195 | 195 | 60 | 440 | 55 | 84.67 | 78.00 | 88.00 | 82.70 |
| CNN + RMSProp | High | 195 | 27 | 28 | 195 | 39 | 461 | 55 | 87.47 | 78.00 | 92.20 | 84.51 |
| | Medium | 20 | 209 | 21 | 209 | 47 | 453 | 41 | 88.27 | 83.60 | 90.60 | 86.96 |
| | Low | 19 | 20 | 211 | 211 | 49 | 451 | 39 | 88.27 | 84.40 | 90.20 | 87.20 |
| CNN + Adam | High | 222 | 12 | 16 | 222 | 23 | 477 | 28 | 93.20 | 88.80 | 95.40 | 91.98 |
| | Medium | 13 | 225 | 12 | 225 | 23 | 477 | 25 | 93.60 | 90.00 | 95.40 | 92.62 |
| | Low | 10 | 11 | 229 | 229 | 28 | 472 | 21 | 93.47 | 91.60 | 94.40 | 92.98 |

Course Label values can be either 1,2 or 3. Courses are categorized as low, medium, and high, based on their complexity level. Value one denotes low, two denotes medium and three denotes high level course.

The starting month and year of the course are given in the course presentation. Assessment type can be in two modes: Quiz or Assignment. For the proposed system to function properly, a sample dataset that includes different parameters such as student ID, course module, durations, number of clicks, gender, *etc*., must be provided as inputs. In the database, there are 125 male and 125 female learners in each category (low, medium, and high), and a total of 750 records are used. Complexity levels of low, medium, and high are denoted as labels 1,2 and 3 respectively.

In a classification system, the "training" phase is used to train the system by making use of labeled data from the "datasets," whereas the "testing" phase is utilized to evaluate the system. The HDL model makes use of $k$-fold cross-validation to partition the database into $k$-folds. The system is evaluated $k$ times using the training data in $k$-1 folds and testing the remaining data.

Then, the overall performance is determined by taking the accuracy of each round and averaging them. The following metrics are used to assess the system's performance: accuracy, precision, recall, and F1-score. To develop a definition of these terms, consideration must be given to the outcomes of the proposed system. Four variables are identified (FP—False Positive, FN—False Negative, TP—True Positive, and TN—True Negative) by counting the number of possible outcomes from the classification task. Then, it is possible to compute the aforementioned performance metrics' namely Accuracy (Acc), Recall (Re), Precision (Pr) and F1 score (F1). The performances of the individual architectures such as CNN, ResNet, and LSTM are analyzed using three optimization techniques such as SGD, RMSProp, and Adam before evaluating the performance of the HDL system. Table 2 shows the CNN-based recommendation system's performances, and its corresponding bar chart for high-, medium- and low-level courses is given in Fig. 6.

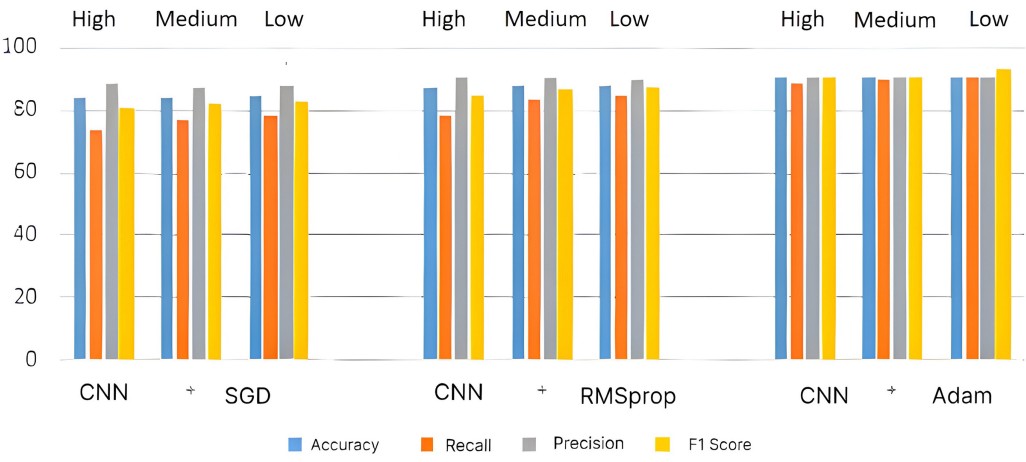

**Figure 6 Performance metrics for CNN architecture.**

It is demonstrated that the CNN + Adam system is capable of achieving a maximum accuracy of ~93%, but the CNN + SGD system is only capable of achieving an accuracy of ~84%. For the same dataset, the CNN + RMSProp system provides better performance than the CNN + SGD system with ~88% accuracy.

To obtain more accurate recommendations, residual units are introduced in the CNN architecture. The bar chart for high-, medium- and low-level courses for the ResNet-based recommendation system's performance is given in Fig. 7. The introduction of residual units increases the performance of the recommendation system by ~1% for course selection. The introduction of residual units provides better performance than CNN-based systems because more important data can be reached to the latter parts of the system by the skipping layers.

To enhance the recommendation, the LSTM architecture is introduced, and the performance metrics are calculated and compared with the other two architectures. The bar chart for the LSTM-based recommendation system's performance metrics is given in Fig. 8. It is inferred that the LSTM-based recommendation system gives more promising results than the CNN- and ResNet-based systems. In the LSTM-based system, the Adam optimizer gives the highest average accuracy of 96.87%, which is 2.35% higher than that of the ResNet system and 3.45% higher than that of the CNN-based system with the same optimization.

To further improve the accuracy of recommended course selection, the aforementioned three architectures are hybridized, and their performances are also evaluated using different optimizers. Table 3 shows the HDL-based recommendation system's performance. It is observed that the HDL system recommends the course for the learner with ~99% accuracy while using the Adam optimizer during training of the three networks such as CNN, ResNet, and LSTM. It is also noted that the performance of the optimizers is in the order of SGD < RMSProp < Adam. Figure 9 shows the performance comparisons of different architectures with different optimizers in terms of average accuracy. From the given dataset, two samples are taken for example to prove the experimental results. Also, the courses are categorized into three levels low, medium, and high, based on their

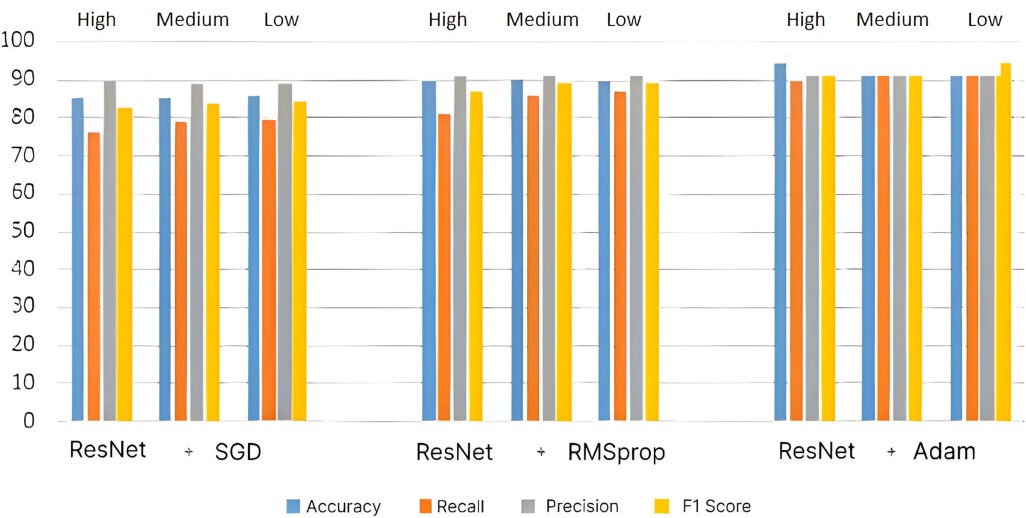

**Figure 7 Performance metrics for ResNet architecture.**

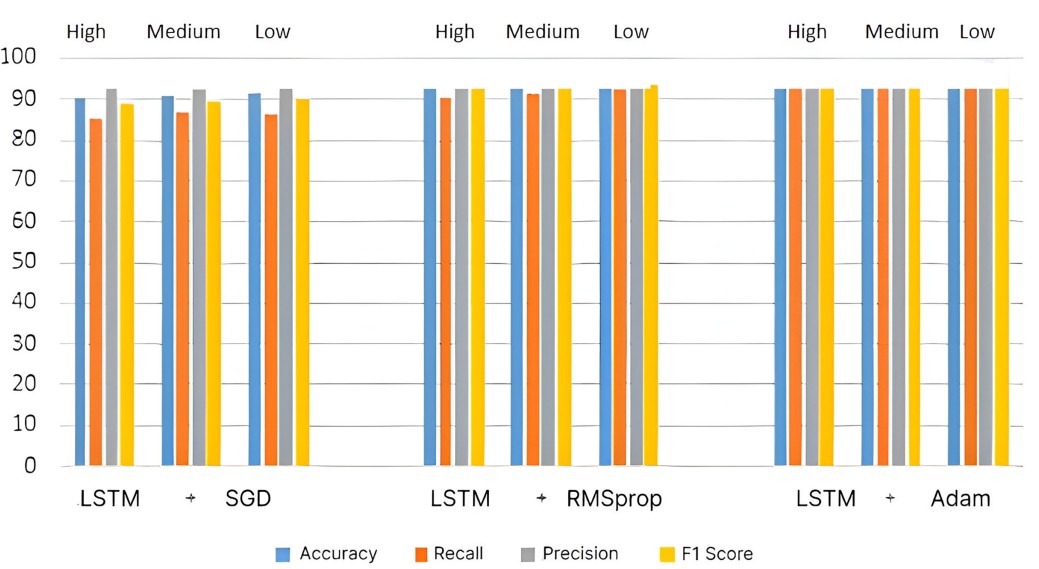

**Figure 8 Performance metrics for LSTM architecture.**

complexity. Complexity level is taken into account based on the course content and the users' state of knowledge. A student with student_id 201801057 has enrolled for the online course "Programming in C" in the Moodle LMS. This course is categorized as a low-level complexity course. He has attempted the quiz and online assignments on time and he has completed the course within the stipulated period of six months. As a next level, the proposed model will recommend him for the next-level course (which is categorized as a medium-level course) on the same stream "Object Oriented Programming". If he regularly submits the assignments and quizzes, he can attempt the final exam. If he successfully clears the exam, the student will again be recommended to do the next-level complexity course "Internet Programming". This course will become easy for him to learn as he has already completed the prerequisites of this subject. Our recommendation system will give

**Table 3 HDL based recommendation system performance.**

| Combination | Confusion matrix | | | | Parameters | | | | Performance indexes | | | |
|---|---|---|---|---|---|---|---|---|---|---|---|---|
| | Category | H | M | L | TP | FP | TN | FN | Acc | Re | Pr | F1 |
| HDL + SGD | High | 235 | 7 | 8 | 235 | 13 | 487 | 15 | 96.27 | 94.00 | 97.40 | 95.67 |
| | Medium | 5 | 239 | 6 | 239 | 13 | 487 | 11 | 96.80 | 95.60 | 97.40 | 96.49 |
| | Low | 8 | 6 | 236 | 236 | 14 | 486 | 14 | 96.27 | 94.40 | 97.20 | 95.78 |
| HDL + RMSProp | High | 242 | 4 | 4 | 242 | 8 | 492 | 8 | 97.87 | 96.80 | 98.40 | 97.59 |
| | Medium | 4 | 243 | 3 | 243 | 7 | 493 | 7 | 98.13 | 97.20 | 98.60 | 97.89 |
| | Low | 4 | 3 | 243 | 243 | 7 | 493 | 7 | 98.13 | 97.20 | 98.60 | 97.89 |
| HDL + Adam | High | 246 | 2 | 2 | 246 | 2 | 498 | 4 | 99.20 | 98.40 | 99.60 | 99.00 |
| | Medium | 1 | 247 | 2 | 247 | 3 | 497 | 3 | 99.20 | 98.80 | 99.40 | 99.10 |
| | Low | 1 | 1 | 248 | 248 | 4 | 496 | 2 | 99.20 | 99.20 | 99.20 | 99.20 |

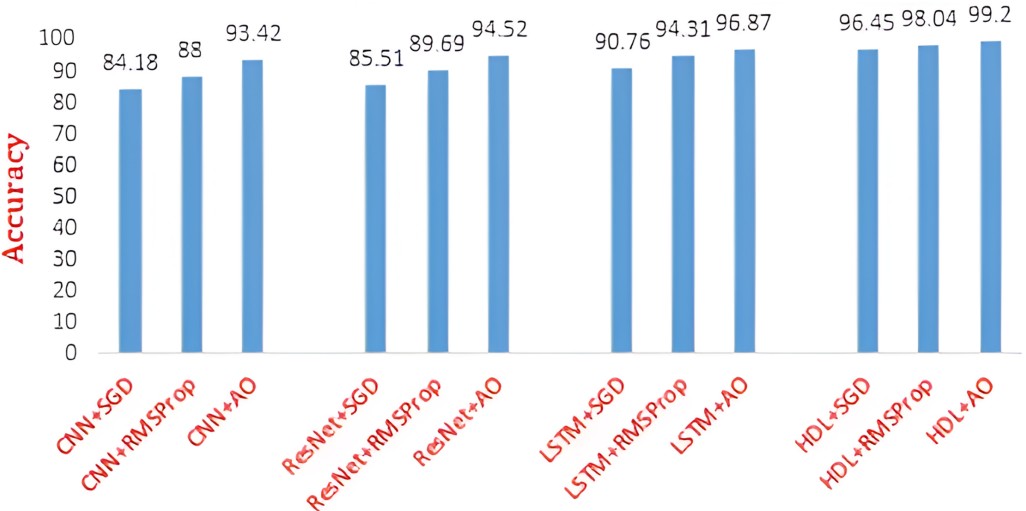

**Figure 9 Performance comparison of different architectures with different optimizers.**

the right direction for the students to choose the courses that fall in the same domain. On the other part, if an enrolled candidate is unable to complete the course on time, he will be recommended to do the same course again within a short period. This may happen due to the late submission of assignments or forgetting to attend the quiz. In these circumstances, warning messages will be sent to the students to alert them to submit assignments/quizzes on time. This factor will motivate the students to complete the course in the future.

## CONCLUSIONS

This model is aimed at designing a framework for learners using deep learning architecture to produce optimal suggestions for course selection. The proposed system, namely, the hybrid deep learning model, may let learners explore all of the available alternative courses

rather than imposing selections on their own. Automatic feature extraction and learning from one-dimensional sequence data may be performed extremely well using the CNN. By including a feedback loop that acts as a form of memory, RNN solves the memory-related issues of CNN. Therefore, the model retains a trace of its previous inputs. The LSTM method expands on this concept by including a long-term memory component in addition to the more conventional short–term memory mechanism. Thus, the combination of CNN and LSTM may provide an effective solution for solving real-time problems such as course selection. In terms of the accuracy of the generated suggestions, the proposed recommendation by the HDL model outperforms the baseline models. It is also noticed that the proposed system improves decision effectiveness, satisfaction, and efficiency. The decision effectiveness has been improved by calculating various metrics namely Precision, Recall, and F1-score. Efficiency can be improved by collecting feedback from the users and thereby enhancing user satisfaction while completing the enrolled course. Most importantly, it provides a much-needed foothold for learners from a vast number of available courses.

## ACKNOWLEDGEMENTS

Our sincere thanks to the Management of Rajalakshmi Institutions, Chennai, Tamil Nadu, India, who are supporting and encouraging us to do the research work. This article is a part of our research work.

### Funding
The authors received no funding for this work.

### Competing Interests
The authors declare that they have no competing interests.

### Author Contributions
- Subha S. conceived and designed the experiments, performed the experiments, analyzed the data, performed the computation work, prepared figures and/or tables, authored or reviewed drafts of the article, and approved the final draft.
- Baghavathi Priya Sankaralingam performed the experiments, analyzed the data, performed the computation work, prepared figures and/or tables, and approved the final draft.
- Anitha Gurusamy performed the experiments, performed the computation work, prepared figures and/or tables, and approved the final draft.
- Sountharrajan Sehar analyzed the data, authored or reviewed drafts of the article, and approved the final draft.
- Durga Prasad Bavirisetti analyzed the data, authored or reviewed drafts of the article, and approved the final draft.

## Data Availability

The data is available at Zenodo: Shuba, & Anitha. (2023). E-Course Recommendation System [Data set]. Zenodo. https://doi.org/10.5281/zenodo.8333444.

## Supplemental Information

Supplemental information for this article can be found online at http://dx.doi.org/10.7717/peerj-cs.1670#supplemental-information.

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
