# Peer review of "Personalization-based deep hybrid E-learning model for online course recommendation system"

_PeerJ Computer Science, doi:10.7717/peerj-cs.1670_

## Round 0.1 · original submission · Major Revisions

Based on the referee reports, I recommend a major revision of the manuscript. The authors should improve the English language, taking carefully into account the comments of the reviewers in the reports and resubmit the paper.

Reviewer 1 ·

Basic reporting

The English language should be improved; thus, proofreading is needed throughout the paper.

The authors should put more emphasis on the importance of course selection for the enrolled candidates on an online learning platform in the introduction.

Most paragraphs in the paper were not appropriately referenced. The paper lacks references; examples include lines 53, 57, 190, 193, 214, 250, 254, 260, etc.

The quality of the figures should be improved.

Experimental design

What is meant by high-, medium-, and low-level courses? please explain

How does the weighted voting method in the probability fusion module work?

If possible, it would be more useful to provide an illustrative example with limited data to explain the steps one-by-one of the proposed model.

Validity of the findings

More details about the dataset should be presented.

Reviewer 2 ·

Basic reporting

This is a potentially interesting paper on a topic that is relevant and useful, however it practical approach is very less as per the recommendations.

Experimental design

Not as per perameters

Validity of the findings

Very Less

Reviewer 3 ·

Basic reporting

All specialists in their respective fields could read and understand it because it was written in formal English that was impartial and objective. This was done to make sure that all professionals could read and comprehend the material. This document contains a sufficient amount of background knowledge as well as pertinent references. The introduction, the summary, and the conclusion are all coherent and accurately present the information. It had proper labeling and a header in addition to the labeling, thus it met all the standards. It encompasses both the methods and the results. The results can be completely recreated, and the code is supplied in a style that makes it easy to execute it. It is able to meet the criteria required for acceptance into the technically appropriate area.

Experimental design

You may feel comfortable submitting it because it is a unique work that fits the "airms and scope" of the journal for which you are writing. The investigation's problem is glaringly evident, so the solution must meet this condition by being both timely and significant. Additionally, the answer closes a gap in the existing body of knowledge; remarks on this aspect of the work's contribution to the literature should be expressed. This closes a gap that existed in the pertinent literature before. It not only complies with the "airms and scope" of the magazine but also with the ethical standards mentioned earlier in this sentence. The steps that must be taken in order to complete an assignment include detailed explanations as well as simulated plans.

Validity of the findings

The findings are presented in a way that is both complete and detailed, and it is original and innovative. The debate is limited to the study results that support the aforementioned topics, and they are connected to the main research questions. Given that both the dataset and the code are accessible to the public and made available for their use, it is easy to duplicate the study's findings. It has been the topic of a review that employs statistical methods and techniques. The advantages of integrating this new corpus of literature in the existing canon have been outlined in a way that is understandable while still being brief.

Additional comments

Figures and tables are displayed towards the end of the article; if they have been inserted where they should be, the piece will have a more professional appearance.

·

Basic reporting

The paper is written in clear and unambiguous English. It provides sufficient literature references, background information, and context for the field. The article follows a professional structure and includes figures, tables, and raw data. However, there are a few areas that could be improved. In the literature review, the author should consider placing more emphasis on ML/DL-based recommendation methods, as the proposed approach is a hybrid deep learning model. While introducing non-ML-based methods is relevant for context, it should be limited to maintain a stronger emphasis on ML/DL-based methods. Additionally, providing a discussion on the pros and cons of existing research works in the field would enhance readers' understanding and allow for a more comprehensive comparison.

Experimental design

The paper presents original primary research within the aims and scope of the journal. The research question is well-defined, relevant, and meaningful, addressing an identified knowledge gap. However, there are some concerns regarding the dataset and experimental design. The number of samples mentioned in the text does not match the number provided in the supplemental files. It is important for the author to provide a detailed description of the dataset format and specify which field the model is predicting. More comprehensive information about the experimental design should also be included.

Validity of the findings

While the underlying data have been provided, there are concerns regarding the validity of the findings due to data issues. The discrepancy between the number of samples mentioned in the text and the number provided in the supplemental files raises questions about the dataset's integrity. The author should address this inconsistency and provide a clear explanation for the discrepancy. Additionally, the lack of a detailed description of the dataset format and the field that the model is predicting hinders the ability to assess the soundness of the experimental design. More comprehensive information about the dataset and experimental design should be included to ensure the robustness of the findings.

The conclusions are well-stated and linked to the original research question, but there are a couple of areas that need revision. The claim that the system can "let learners explore all of the available alternative courses rather than imposing selections on their own" is inaccurate and should be revised to reflect the limitations of the system. Additionally, the claim regarding the improvement of decision effectiveness, satisfaction, and efficiency lacks justification or evidence throughout the paper and should be revised to accurately reflect the scope and limitations of the study.

Additional comments

Literature reference:
1. The author should consider placing more emphasis on ML/DL-based recommendation methods in the paper, given the proposed approach is a hybrid deep learning model. While introducing non-ML-based recommendation methods is relevant to provide context, it is advisable for the author to limit the extent of their coverage. This would ensure that the paper maintains a stronger emphasis on ML/DL-based methods, aligning with the main objectives and contributions of the study.

2. It would be beneficial if the author could provide a discussion on the pros and cons of the existing research works in the field. This would enhance the understanding of the readers by providing insights into the strengths and weaknesses of different approaches, allowing for a more comprehensive comparison.

Model design:
1. Line 194-197: Please provide further justification for assuming that the data does not follow a Gaussian distribution. It would be helpful to include statistical metrics or a figure illustrating the distribution to support this claim. This would strengthen the validity of your analysis and enhance the readers' understanding of the underlying data distribution. Overall, the reasons for favoring one deep learning model over others should be thoroughly reconsidered and discussed in a more nuanced manner.

2. Line 248-254 and Line 287-294: The claim that the primary advantage of using CNN is reduced overfitting should be reevaluated. Similarly, the statement that the preference for ResNet over CNN is mainly due to the amount of data should be reconsidered. It is recommended to explore and discuss other factors, such as network depth, skip connections, task-specific performance, or architectural innovations, that may contribute to the preference for ResNet. Overall, the reasons for favoring one deep learning model over others should be thoroughly reconsidered and discussed in a more nuanced manner.

3. Line 334-342: Please provide a concise explanation for why the Softmax function was exclusively chosen for the eLearning recommendation task, rather than considering other activation functions like Sigmoid or ReLU.

4. Line 364-366: It is crucial to clarify how the probabilities from each separate model are combined, as this represents the key point of the entire paper, which proposes a hybrid deep learning model. Providing a detailed explanation of the methodology or algorithm used to combine these probabilities would significantly enhance the readers' understanding and reinforce the significance of the proposed approach.

Experiments:
1. Line 430-432: The author states that their dataset comprises 125 male and 125 female samples, with a total of 750 samples. However, upon reviewing their supplemental files, it is observed that only 200 records are provided.

2. The author should provide a detailed description of their dataset format, including information on the input of their models and how each data field is processed. Additionally, it is important for the author to specify which field their model is predicting. Furthermore, more comprehensive information about the experimental design should be included, explaining key details of how the experiment was conducted.

Figures:
1. Figure 1: It is important for the authors to describe the data preprocessing steps in their text, as these details are currently missing. Additionally, there appears to be an inconsistency between the information provided in the text and the figure regarding data normalization/standardization. According to the text, data standardization is not utilized in the system, but it is depicted in the figure.

3. Figure 7-9: It is recommended to consider adjusting the scale of the y-axis. As all the metrics in the figures never exceed 100, the maximum value of the y-axis should be set accordingly and not exceed 100.

4. Figure 7-9: Add the label for high, medium and low level courses.

5. In general, the figures would benefit from several improvements. Consider resizing and rotating the figures as needed to enhance readability and clarity. Additionally, adjusting the font size to ensure legibility is also recommended. Finally, pay attention to better alignment of elements within the figures.

Conclusion:
1. Line 487-489: Based on the findings from their experiments, it is appropriate to conclude that the model is capable of predicting the courses learners selected. However, it would be inaccurate and overly assertive to claim that the system can "let learners explore all of the available alternative courses rather than imposing selections on their own." It is recommended to revise the statement to accurately reflect the limitations of the system and the specific capabilities demonstrated through the experiments. This would ensure a more accurate and measured conclusion based on the experimental results.

2. Line 497-499: The claim made by the author regarding the improvement of decision effectiveness, satisfaction, and efficiency in the proposed system appears overly aggressive. Throughout the paper, there is a lack of justification or evidence provided to support the assertion of improved learner satisfaction, decision effectiveness, or efficiency. It is recommended to revise this claim to accurately reflect the scope of the study and the limitations in assessing these factors.

3. Line 491: typo -- feedback "loop"

---

## Round 0.2 · Major Revisions

Based on the reviewer comments I am suggesting you revise the manuscript and resubmit.

**Language Note:** The review process has identified that the English language must be improved. PeerJ can provide language editing services - please contact us at copyediting@peerj.com for pricing (be sure to provide your manuscript number and title). Alternatively, you should make your own arrangements to improve the language quality and provide details in your response letter. – PeerJ Staff

Reviewer 1 ·

Basic reporting

no comment

Experimental design

no comment

Validity of the findings

no comment

Additional comments

I'm pleased to inform you that the authors have addressed my comments, enhancing the paper's quality. The revisions have contributed positively to its overall presentation and depth.

·

Basic reporting

No new comments on this section in the second submission. Previous comments were unchanged.

Experimental design

1. Model Design:
- Line 332-341: The reason why the data does not follow a Gaussian distribution remains unclear. This issue was raised in the first review and remains unaddressed in the second submission.
- Line 470-480: No changes were observed in these lines compared to the previous version, despite the suggestions provided in the initial review.
- Line 596-613: No changes were made regarding the justification for the exclusive use of the Softmax function. The first review explicitly requested further explanation on this.
- Line 656-660: No changes or clarifications were made about how the probabilities from each separate model are combined, which is a key aspect of the paper. This concern was previously raised and remains unaddressed.

2. Experiments:
- Line 822-825: The discrepancy in the number of samples mentioned remains unchanged. This inconsistency was highlighted in the first review but was not addressed in the second submission.

Validity of the findings

Given the unchanged experimental design and unresolved issues, concerns remain regarding the validity of the findings. The discrepancies and lack of clarity can potentially hinder the robustness of the findings. The author needs to resolve these issues to ensure the soundness of the research.

Additional comments

1. The second submission appears to have minimal changes, with many critical feedback points from the first review unaddressed. It's concerning that multiple significant suggestions were not considered or incorporated into the revised manuscript.

2. The writing and presentation could benefit from further refinement. Improving the clarity, organization, and emphasis on critical points would make the paper more comprehensible and impactful.

3. The unchanged nature of the manuscript suggests a lack of engagement with the review process, which could potentially hinder the paper's contribution and significance to the field.

---

## Round 0.3 · Minor Revisions

Kindly revise the manuscript as per the reviewer suggestions and resubmit it.

·

Basic reporting

The authors have presented the manuscript in a clear and largely unambiguous manner. The use of professional English throughout facilitates comprehension for the readers. The literature references and the background provided serve as a solid foundation for understanding the context of the research. This adds depth to the paper and shows the groundwork done by the authors. The structure of the article aligns with professional standards, with well-presented figures and tables. The provision of raw data would enhance the transparency and reproducibility of the research. The results are presented in a self-contained manner, making it straightforward for readers to relate the findings to the stated hypotheses.

Experimental design

The manuscript focuses on original primary research that aligns with the aims and scope of the journal. The research question is well defined, and the authors have made an effort to highlight the knowledge gap they aim to address. However, there remains room for clarity in certain areas. Overall, the investigation appears to be rigorous and adheres to both technical and ethical standards. It would be beneficial for the research community if some methods, like the combination of probabilities from different models, were described in more detail. Replication of this work would benefit from a slightly more detailed methodology, ensuring other researchers can understand and replicate the process faithfully.

Validity of the findings

The underlying data presented in the manuscript appears to be consistent and statistically sound. However, some discrepancies, such as in the number of samples mentioned, should be reconciled to strengthen the paper's validity. The conclusions provided are logically derived from the findings. Some clarity on unresolved issues would further cement the validity of these conclusions.

Additional comments

The manuscript shows promise and holds value for the academic community. Addressing the few remaining concerns, mainly in the experimental design, would elevate the paper's contribution to the field. The manuscript's overall presentation would benefit from additional refinements in clarity and organization.

---

## Round 0.4 · accepted · Accept

The author has addressed the reviewers' comments properly. Thus I recommend publication of the manuscript.